# Neural network aided approximation and parameter inference of non-Markovian models of gene expression

Qingchao Jiang [1,5], Xiaoming Fu [1,2,5], Shifu Yan[1,5], Runlai Li[3], Wenli Du[1], Zhixing Cao [1,4✉], Feng Qian[1] & Ramon Grima [2✉]

Non-Markovian models of stochastic biochemical kinetics often incorporate explicit time delays to effectively model large numbers of intermediate biochemical processes. Analysis and simulation of these models, as well as the inference of their parameters from data, are fraught with difficulties because the dynamics depends on the system's history. Here we use an artificial neural network to approximate the time-dependent distributions of non-Markovian models by the solutions of much simpler time-inhomogeneous Markovian models; the approximation does not increase the dimensionality of the model and simultaneously leads to inference of the kinetic parameters. The training of the neural network uses a relatively small set of noisy measurements generated by experimental data or stochastic simulations of the non-Markovian model. We show using a variety of models, where the delays stem from transcriptional processes and feedback control, that the Markovian models learnt by the neural network accurately reflect the stochastic dynamics across parameter space.

[1] Key Laboratory of Smart Manufacturing in Energy Chemical Process, Ministry of Education, East China University of Science and Technology, Shanghai, China. [2] School of Biological Sciences, The University of Edinburgh, Edinburgh, Scotland, UK. [3] Department of Chemistry, National University of Singapore, Singapore, Singapore. [4] State Key Laboratory of Bioreactor Engineering, East China University of Science and Technology, Shanghai, China. [5] These authors contributed equally: Qingchao Jiang, Xiaoming Fu, Shifu Yan. ✉email: zcao@ecust.edu.cn; ramon.grima@ed.ac.uk

Over the past two decades, stochastic modelling has provided insight into how cellular dynamics is influenced by noise in gene expression[1–3]. The complexity of cellular biochemistry prevents a full stochastic description of all reaction events and rather these models are effective, in the sense that each reaction provides an effective description of a group of processes. A major assumption behind the majority of stochastic models of biochemical kinetics is the memoryless hypothesis, i.e., the stochastic dynamics of the reactants is only influenced by the current state of the system, which implies that the waiting times for reaction events obey exponential distributions. In the context of stochastic gene expression, the telegraph model (or the two-state model)[4–6] is a fundamental Markovian model describing promoter switching, transcription and degradation of mature RNA. While this Markovian assumption considerably simplifies model analysis[7], it is dubious for modelling certain non-elementary reaction events that encapsulate multiple intermediate reaction steps. For example, consider a model of transcription that predicts the distribution of RNA polymerase (RNAP) numbers attached to the gene but which does not explicitly model the microscopic processes behind elongation[8]. In this case, assuming that RNA synthesis proceeds with approximately constant elongation speed, the reaction modelling termination should occur a fixed time after the reaction modelling initiation, which implies that the system has memory and is not Markovian. Of course in this instance, the model could be made Markovian by extending it so that it includes the explicit microscopic description of the movement of the RNAP along the gene[9] but this implies a large increase in the effective number of species, which makes explicit solution of the Markovian model impossible. Hence in many cases, a low dimensional stochastic model can be only achieved by a suitable non-Markovian description. Given their practical importance, these systems have been the subject of increased research interest, leading to an exact analytical solution for a few simple cases and the development of exact Monte Carlo algorithms for the study of those with complex dynamics[8,10–16].

Nevertheless, presently the understanding of non-Markovian models lags much behind that of Markovian models where a wide range of approximation methods are available. Hence there is ample scope for the development of methods to tackle the difficulties inherent in stochastic systems possessing memory. Given the success of artificial neural networks (ANNs) in solving scientific problems where traditional methods have made little progress, it is of interest to consider whether such an approach could be useful for solving the aforementioned stochastic problems. Neural networks being universal function approximators have recently been used to solve partial differential equations commonly used in physics, biology and chemistry. In particular these techniques have been used to approximately solve Burgers equation[17–19], the Schrodinger equation[18,20] and partial differential equations describing collective cell migration in scratch assays[21]; the ANN-based methods behind the solution of these problems are a subclass of the universal differential equation framework that has recently been proposed[22].

Inspired by the success of ANNs in other fields, in this article we develop a novel ANN-based methodology to study non-Markovian models of gene expression and transcriptional feedback. We propose to use an ANN to approximate non-Markovian models by much simpler stochastic models. Specifically the key idea is to approximate the chemical master equation (CME) of non-Markovian models (which we refer to as the delay CME) that is in terms of the two-time probability distribution by a CME whose terms are only a function of the current time, i.e. by a time-inhomogeneous Markov process (see Fig. 1a for an illustration). Notably, this mapping is achieved without increasing the number of fluctuating species. We refer to the learnt CME describing the time-inhomogeneous Markov process as the neural-network chemical master equation (NN-CME). The latter, because of its simplified form, can then either be studied analytically using standard methods or else straightforwardly simulated using the finite state projection (FSP) method. In what follows, we introduce the ANN-based approximation method by means of a simple example and then verify its accuracy in predicting time-dependent molecule number distributions of various realistic models of gene expression and its superior computational efficiency when compared to direct stochastic simulation. We finish by showing the orthogonal use of the method to infer the parameters of bursty gene expression from synthetic data.

## Results

**Illustration of ANN-aided stochastic model approximation using a simple model of transcription**. We consider a simple non-Markovian system where molecules are produced at a rate $\rho$ and are removed from the system (degraded) after a fixed time delay $\tau$:

$$\varnothing \xrightarrow{\rho} N, \quad N \underset{\tau}{\Rightarrow} \varnothing. \tag{1}$$

In other words, each molecule has an internal clock that starts ticking when it is 'born', and when this clock registers a time $\tau$, the molecule 'dies'. Note that as a convention in this paper, we use an arrow to denote a reaction in which the products are formed after an exponentially distributed time and an arrow with two horizontal lines to denote a reaction, which occurs after a fixed elapsed time. The above model, which we denote Model I, describes the fluctuations of nascent RNA ($N$) numbers due to constitutive expression. Specifically, the production reaction models the process of initiation whereby an RNAP molecule binds the promoter; the delayed-degradation reaction models, in a combined manner, the processes of elongation and termination whereby an RNAP molecule travels at a constant velocity along the gene and finally detaches from the gene, respectively. Note that the number of RNAPs bound to the gene is equal to the number of nascent RNA molecules present, irrespective of their lengths[23] (for an illustration see Fig. 2a Model I). We note that the signal from single-molecule fluorescence in situ hybridization (smFISH) probes corresponds to measuring the total length of nascent RNA, summed over multiple molecules present at the gene; thus the number of nascent RNA estimated from such experiments leads to a continuous rather than a discrete number[8,24,25]. Our present formulation ignores the complexities introduced by smFISH and is rather compatible with experiments that can directly quantify the number of RNAPs bound to a gene[26].

It can be shown (see SI Note 1) from first principles that the delay CME describing the stochastic dynamics of this process is given by:

$$\frac{dP(n, t)}{dt} = \rho[P(n-1, t) - P(n, t)] + \rho[P(n, t|0, t-\tau) \\ - P(n-1, t|0, t-\tau)], \tag{2}$$

where $P(n, t)$ is the probability that at time $t$ there are $n$ nascent RNA molecules bound to the gene. Similarly $P(n, t|n', t')$ is the conditional probability distribution that at time $t$ there are $n$ molecules given that at a previous time $t'$, there were $n'$ molecules. The right hand side of the master equation is a function of the present time $t$ as well as of the previous time $t - \tau$. Master equations of this type are typically much more difficult to solve, analytically or numerically, than conventional master equations where the right hand side is only a function of the

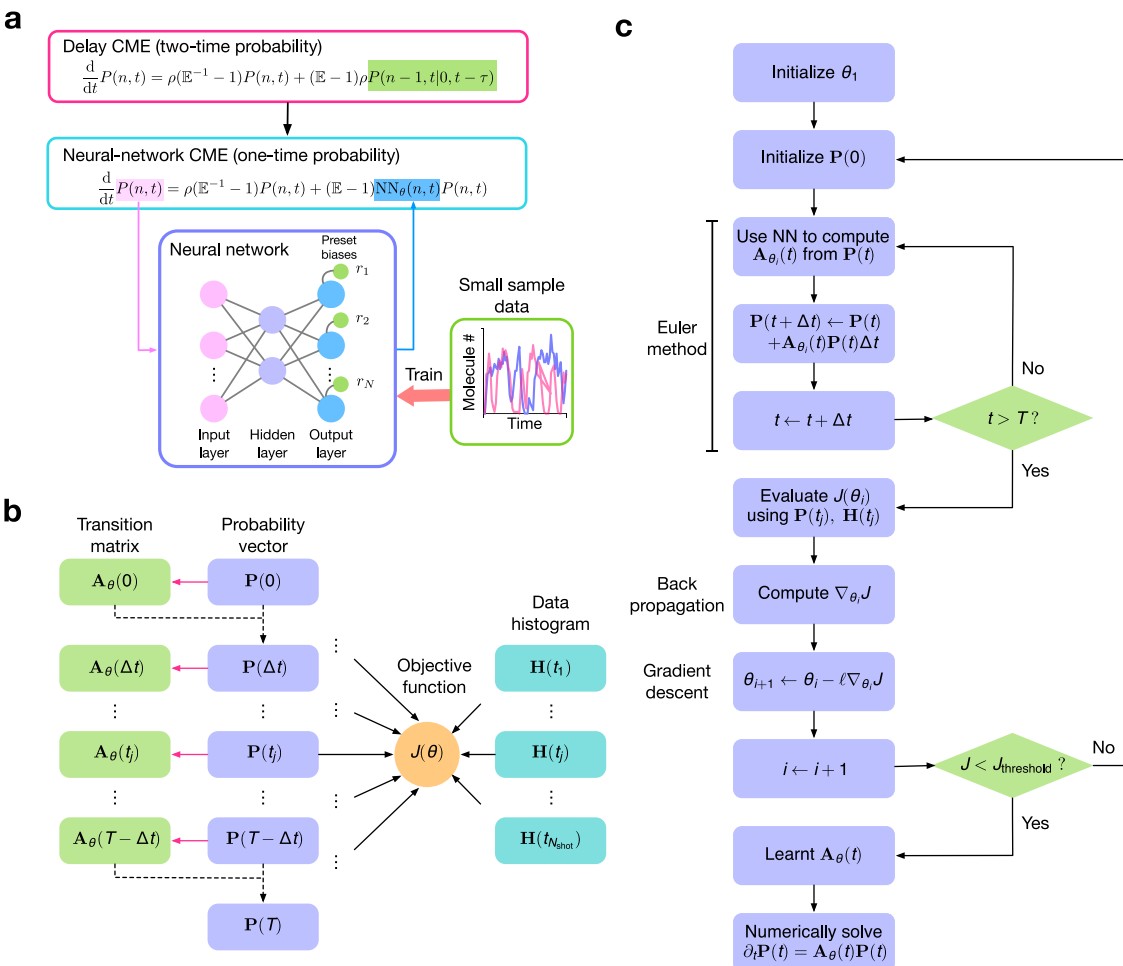

**Fig. 1 The ANN-aided stochastic model approximation. a** Illustration of the key idea behind the method, namely the ANN-aided mapping of a delay master equation that is in terms of the two-time probability distribution by the simpler neural-network chemical master equation (NN-CME) whose terms are only a function of the current time. **b** Illustration of the procedure behind the calculation of the transition matrix and the objective function. For a given set of weights and biases of the ANN (denoted by $\theta$), taking as input $\mathbf{P}(t)$, the ANN's output gives the transition matrix elements $\mathbf{A}_\theta(t)$, which then by means of the Euler method (or more advanced differential equation solvers) is used to predict the distribution at the next time step $\mathbf{P}(t + \Delta t)$. Note that magenta arrows show the ANN computation while the black dashed arrows show the use of the Euler method. Stochastic simulations that sample the solution of the delay master equation are used to produce histograms at several time points $\mathbf{H}(t)$; finally the distance $J(\theta)$ is calculated between the latter and $\mathbf{P}(t)$ (evaluated at the same time points). **c** Flowchart illustrating all the steps in ANN training. If the objective function calculated as shown in (**b**) is above a threshold then the weights and biases of the ANN are updated using back propagation followed by gradient descent; this is repeated until the objective function is below a threshold.

present time $t$ (because of its simplicity an exact time-dependent solution of Model I is possible and shown in SI Note 1; see also[12]). Hence the key idea of our method is to map the master Eq. (2) to the new master Eq. (3):

$$\frac{d}{dt}P(n, t) = \rho[P(n - 1, t) - P(n, t)]$$
$$+ \mathrm{NN}_\theta(n + 1, t)P(n + 1, t) - \mathrm{NN}_\theta(n, t)P(n, t),$$
$$(3)$$

where the function $\mathrm{NN}_\theta(n, t)$ is an effective time-dependent propensity describing the removal of nascent RNA molecules, which is to be learnt by the ANN. This is the NN-CME for reaction scheme (1). Note that this master equation with $\mathrm{NN}_\theta(n, t) = kn$ is the conventional CME describing the birth–death process $\varnothing \xrightarrow{\rho} N, N \xrightarrow{k} \varnothing$, where $k$ is the degradation rate. By considering the cases $n = 0, ..., N$ of Eq. (3) (where $N$ is some positive integer much larger than 1), one obtains a system of $N + 1$ differential equations. These equations need to

be closed before they can be solved. First, we can set $P(-1)$ $(t) = 0$ since the number of nascent RNA cannot become negative. Next, since we have truncated space to $n = N$, it follows that any terms in the equations corresponding to jumps from $n = N$ to $n = N + 1$ or vice versa, need to be neglected. This implies that the terms $-\rho P(N, t)$ and $\mathrm{NN}_\theta(N + 1, t)P(N + 1, t)$ are neglected. This is indeed the main idea behind FSP[27], which leads to a finite closed system of differential equations for the probabilities. Of course to faithfully approximate the system's dynamics, $N$ should be chosen large enough such that $P(N, t) \approx 0$; in practice $N$ is chosen such that any further increase in its value leads to no significant change in the solution of the master equation. The closed system of equations can be compactly written in the form

$$\frac{d}{dt}\mathbf{P}(t) = \mathbf{A}_\theta(t)\mathbf{P}(t), \qquad (4)$$

with $\mathbf{P}(t) = [P(0, t), ..., P(N, t)]^\top$. The transition matrix is defined as $\mathbf{A}_\theta = \mathbf{D} + \mathbf{N}_\theta(t)$, where the two components

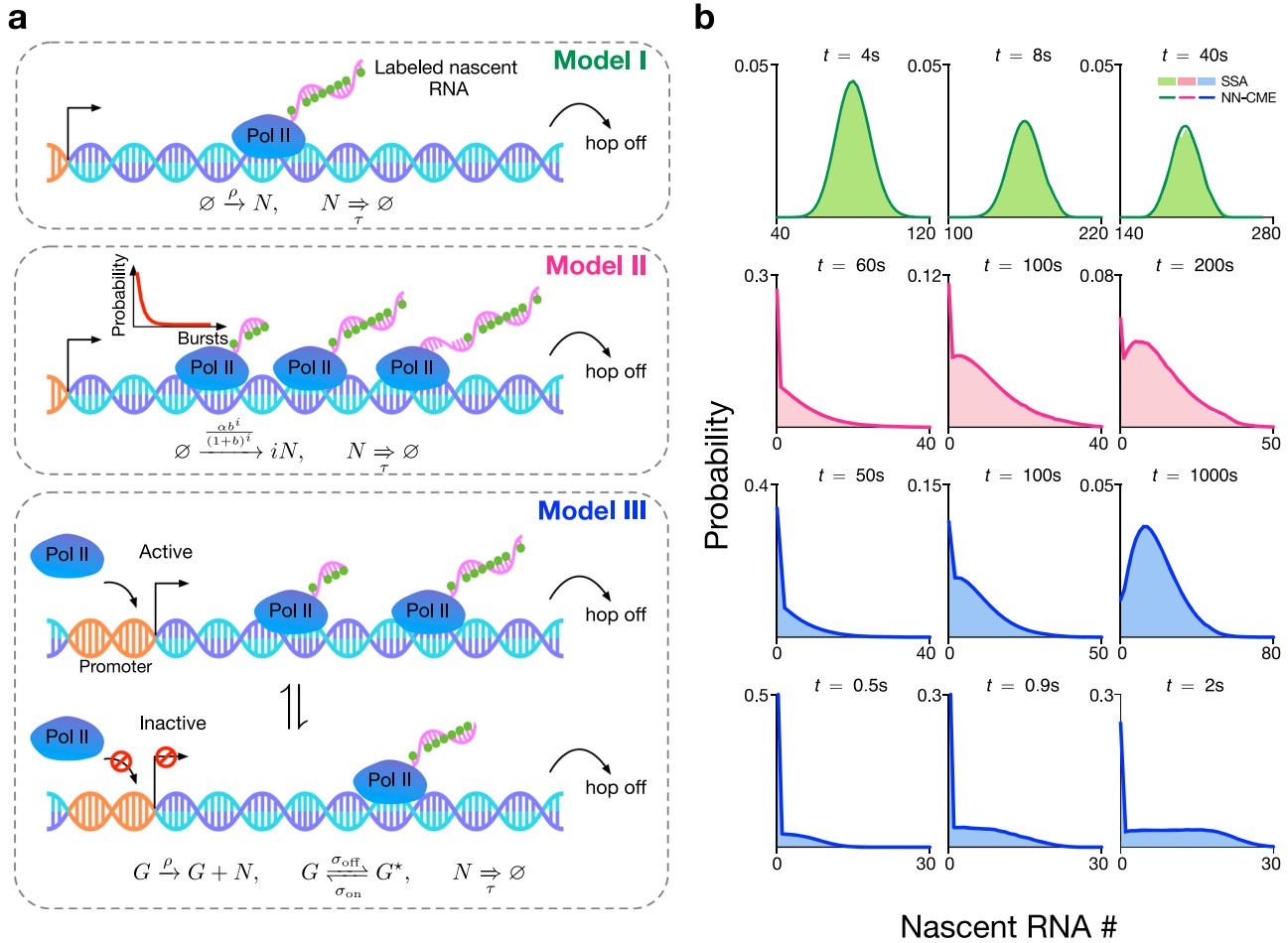

**Fig. 2 ANN-aided stochastic model approximation of various models of transcription. a** Illustration of three models of transcription. The models describe initiation, elongation and termination and specifically predict the numbers of nascent RNAs (equivalently the number of RNAP polymerases, Pol IIs) at the gene locus. In all models, a nascent RNA molecule detaches after a constant time has elapsed from its binding to the promoter. The models differ in how they model Pol II binding: in Model I, the binding is modelled as a Poisson process, hence one at a time; in Model II, binding occurs in bursts, whose size conforms to a geometric distribution; in Model III, the gene switches between active and inactive states, and only the active state permits Pol II binding. **b** For all models, the FSP solution of the NN-CME derived by the ANN-aided procedure is in excellent agreement with the SSA of the delay CME. The accuracy is independent of the modality and skewness of the distribution. The rate constants and other parameters related to the ANN's training are specified in SI Table 1.

are given by

$$
\mathbf{D} = \begin{bmatrix}
-\rho & 0 & \cdots & 0 & 0 \\
\rho & -\rho & \cdots & 0 & 0 \\
\vdots & \vdots & \ddots & \vdots & \vdots \\
0 & 0 & \cdots & -\rho & 0 \\
0 & 0 & \cdots & \rho & 0
\end{bmatrix},
$$

and

$$
\mathbf{N}_\theta(t) = \begin{bmatrix}
0 & \mathrm{NN}_\theta(1,t) & \cdots & 0 & 0 \\
0 & -\mathrm{NN}_\theta(1,t) & \cdots & 0 & 0 \\
\vdots & \vdots & \ddots & \vdots & \vdots \\
0 & 0 & \cdots & -\mathrm{NN}_\theta(N-1,t) & \mathrm{NN}_\theta(N,t) \\
0 & 0 & \cdots & 0 & -\mathrm{NN}_\theta(N,t)
\end{bmatrix}.
$$

The output $\mathrm{NN}_\theta(0,t)$ is set to 0 to reflect the fact that nascent RNA cannot be further removed when there is none attached to the gene.

Next we describe how we train the ANN to infer the effective transition matrix $\mathbf{A}_\theta(t)$. We use a multilayer perceptron with a single hidden layer; this is a simple feedforward ANN consisting of three layers—an input layer with $N+1$ inputs, a hidden layer with an arbitrary number of neurons and an output layer with $N$ outputs. The simplicity of the ANN here used is motivated by the universal approximation theorem, which states that a single hidden-layer feedforward ANN is able to approximate a wide class of functions on compact subsets[28,29]. The activation functions used in the hidden layer and output layer are `tanh` and `relu` for all examples. In the output layer, we impose an increasing set of fixed biases, which we specify later. For more details of the ANN, including the choice of hyperparameters, please see SI Table 1. The training procedure consists of three main steps:

(1) We use stochastic simulations of the birth delayed-degradation reaction (1) to generate approximate probability distributions at the time points $t \in [t_1, t_2, ..., t_{N_\mathrm{shots}}]$, where $N_\mathrm{shots}$ is the total number of snapshots. Note that by stochastic simulations in

this paper, we always mean an exact stochastic simulation algorithm (SSA) modified to incorporate delays (specifically Algorithm 2 in ref. [10]; for proof of its exactness see ref. [11]). Let these distributions be denoted as $\mathbf{H}(t)$.

(2) The initial condition $\mathbf{P}(0)$ is set to be the same as $\mathbf{H}(0)$. The $N + 1$ elements of this probability vector constitute the inputs to the ANN. Given some set of weights and biases $\theta$, the ANN's $N$ outputs are then taken as the elements of the matrix $\mathbf{N}_\theta(0)$, i.e. the $n$th output of the ANN is $NN_\theta(n, 0)$. By a numerical discretization of Eq. (4), given the inputs and the outputs of the ANN, we obtain $\mathbf{P}(\Delta t)$, where $\Delta t$ is the finite time step. This procedure can be iterated to obtain $\mathbf{P}(2\Delta t)$, $\mathbf{P}(3\Delta t)$, etc. Hence we obtain the probability vector $\mathbf{P}(t)$ at the time points $t \in [t_1, t_2, ..., t_{N_{\text{shots}}}]$.

(3) We calculate an objective function that is a measure of the distance between the distributions $\mathbf{H}(t)$ and $\mathbf{P}(t)$ summed over all snapshots. If the objective function is larger than a threshold then update the set of weights and biases by means of back propagation and gradient descent, and repeat step 2. If the distance is smaller than the threshold then the procedure ends and the transition matrix $\mathbf{A}_\theta(t)$ has been learnt.

Note that since the output of the ANN is the propensities $NN_\theta(n, t)$, these must be positive. We choose the set of biases in the output layer ($r_n$ in Fig. 1a for $n = 1, ..., N$) to be given by $r_n = \frac{n}{\tau}$. This form is inspired by the fact that for the conventional CME with first-order degradation, $NN_\theta(n, t)$ is proportional to $n$, where the proportionality constant is the effective rate of decay, which has units of inverse time. Hence intuitively, the effective removal propensity of the NN-CME is equal to the propensity assuming first-order degradation plus a correction, which is what the ANN effectively learns. This choice of biases is also similar to that of well-known residual network (ResNet)[30,31], which helps to accelerate the convergence of training and reduce computational cost.

Note also that the objective function is chosen as $J(\theta) = \sum_{j=1}^{N_{\text{shot}}} \| \mathbf{H}(t_j) - \mathbf{P}(t_j) \|_2^2$. While there are more accurate distance measures (such as the Wasserstein distance), we use the mean-squared-error form for two reasons: (i) it is commonly used in neural-network training, and (ii) its simple form allows efficient calculation of derivatives through the chain rule (the back propagation method). The steps of the training procedure are illustrated in Fig. 1b, c. Note that while the gradient descent in Fig. 1c is illustrated using an Euler method, for our training we used the standard adaptive moment estimation algorithm (ADAM).

Once the matrix $\mathbf{A}_\theta(t)$ is learnt, Eq. (4) can be integrated numerically to obtain the time-dependent probability vector at all times in the future. In Fig. 2b (first row), we show that the solution of the learnt NN-CME is practically indistinguishable from distributions estimated from stochastic simulations of Model I (1)—hence this implies that the ANN training protocol is effective as a means to map a master equation with terms having a non-local temporal dependence to a master equation with terms having a purely local temporal dependence.

**Testing the accuracy and computational efficiency of ANN-aided stochastic model approximation on more complex models of transcription**. To verify that the accurate mapping characteristics are not specific to Model I, we next consider the application of the procedure to learn the NN-CME of two more complex transcription models incorporating delay (see Fig. 2a). We consider Model II, which is the same as Model I, except that the binding of RNAPs to the promoter occurs in bursts whose

size $i$ is distributed according to the geometric distribution $b^i/(1 + b)^{i+1}$; this can be described by the reaction scheme:

$$\varnothing \xrightarrow{\frac{\alpha b^i}{(1+b)^{i+1}}} iN, \ i = 0, 1, 2, ... \tag{5}$$
$$N \underset{\tau}{\Rightarrow} \varnothing,$$

where $\alpha$ stands for the burst frequency and $b$ is the mean burst size. This is a minimal delay model to describe the phenomenon of transcriptional bursting[32]. The delay CME describing the nascent RNA dynamics is given by (see SI Note 2):

$$\partial_t P(n, t) = \sum_{m=1}^{\infty} \frac{\alpha b^m}{(1 + b)^{m+1}} [P(n - m, t) - P(n, t)]$$
$$+ \sum_{m=1}^{\infty} \frac{\alpha b^m}{(1 + b)^{m+1}} \sum_{n'} P(n', t)[P(n + m, t | n' + m, t - \tau)$$
$$- P(n, t | n' + m, t - \tau)]. \tag{6}$$

This equation can be solved analytically for the time-dependent probability distribution (see SI Note 2).

We also consider Model III wherein the promoter switches between an active and inactive state, RNAP binding occurs only in the active state, which is followed by delayed degradation modelling the RNAP movement along the gene and its detachment; this can be described by the reaction scheme:

$$G \xrightarrow{\rho} G + N, \quad G \underset{\sigma_{\text{on}}}{\overset{\sigma_{\text{off}}}{\rightleftharpoons}} G^\star, \quad N \underset{\tau}{\Rightarrow} \varnothing, \tag{7}$$

where $G$ and $G^\star$ stand for the active and inactive gene state, respectively, and $\sigma_{\text{on}}$ and $\sigma_{\text{off}}$ are the activation and inactivation rates, respectively. It can be shown that in the limit of large $\sigma_{\text{off}}$ (compared to $\sigma_{\text{on}}$), Model III reduces to Model II, whereas in the opposite limit of small $\sigma_{\text{off}}$, it reduces to Model I. Hence Model III can describe both constitutive and bursty transcription, as well as regimes in between. The delay CME describing nascent RNA dynamics is given by (see SI Note 3):

$$\frac{d P_0(n, t)}{d t} = -\sigma_{\text{on}} P_0(n, t) + \sigma_{\text{off}} P_1(n, t) + \sum_{n'} \rho P_1(n', t - \tau)$$
$$[P_{01}(n, t | 0, t - \tau) - P_{01}(n - 1, t | 0, t - \tau)],$$
$$\frac{d P_1(n, t)}{d t} = \sigma_{\text{on}} P_0(n, t) - \sigma_{\text{off}} P_1(n, t) \tag{8}$$
$$+ \rho[P_1(n - 1, t) - P_1(n, t)] + \sum_{n'} \rho P_1(n', t - \tau)$$
$$[P_{11}(n, t | 0, t - \tau) - P_{11}(n - 1, t | 0, t - \tau)],$$

where $P_i(n, t)$ is the probability that the gene is in state $i$ at time $t$ and the number of nascent RNA is $n$; note that $i = 0, 1$ where 0 is the inactive state and 1 is the active state. Similarly $P_{ij}(n, t | n', t')$ denotes the conditional probability distribution that at time $t$ the gene is in state $i$ and the number of molecules is $n$, given that at a previous time $t'$, the gene was in state $j$ and the number of molecules was $n'$. We note that an exact closed-form solution for the steady-state distribution of this model was reported in ref. [8]. The method involves writing the time-evolution equation for the characteristic function and solving it explicitly by means of the Dyson series. Solutions are in fact also possible if the model is modified to predict the signal from smFISH, which necessitates the use of continuous rather than discrete nascent RNA numbers.

Note that as for the delay CME describing Model I, the delay CMEs describing Models II and III also have terms on the right hand side, which are a function of the two-time probability distribution. These terms which stem from delayed degradation, make analytical and numerical solution of the delay master

equations non-trivial. However the ANN-aided procedure to solve Models II and III is as easy to implement as for Model I. By replacing the two-time probability distribution terms on the left-hand sides of Eqs. (6) and (8) by terms of the type $NN_\theta(n, t)$ (see SI Note 4 for details), one can map the delay master equations into NN-CMEs of the form $\frac{d}{dt}\mathbf{P}(t) = \mathbf{A}_\theta(t)\mathbf{P}(t)$, where the transition matrix $\mathbf{A}_\theta(t)$ is learnt by the same training procedure as before. Note that the NN-CME for Model III is none other than the telegraph model of gene expression[4] but modified to allow degradation propensities to be some general function of nascent copy number and time, and specific to each promoter state.

In Fig. 2b rows 2, 3 and 4, we show the comparison between the time-dependent distribution of nascent RNA predicted by the NN-CME and stochastic simulations of the reaction schemes corresponding to Models II and III. The agreement is excellent at all times and for all models, independent of the modality and skewness of the distribution. This reinforces the result that the ANN-aided procedure enables an accurate mapping of master equations with terms having a non-local temporal dependence (via the two-time probability distribution) to master equations with terms having a purely local temporal dependence.

Next, we test the computational efficiency of the ANN-aided procedure compared to stochastic simulations. Figure 3a shows the Hellinger distance between the probability distribution of nascent RNA numbers according to the NN-CME and the exact analytical solution of Model I (see SI Note 1) as a function of the number of snapshots $N_{shots}$ and of the number of stochastic simulations used to train the ANN. As expected, increasing the number of snapshots and the number of stochastic simulations in the ANN's training enhances the accuracy of the NN-CME's distribution (manifested as a reduction in the Hellinger distance). More interestingly, we found that the NN-CME obtained from training the ANN with just a thousand stochastic simulations outputs a distribution that has the same Hellinger distance from the exact distribution as the one obtained from a histogram generated using 30,000 stochastic simulations (direct simulation). Moreover in this case, the time-to-acquire samples plus the time for ANN training takes 1/6 of the computation time if we only use simulations. Another way of distinguishing our method and stochastic simulations is to compare the distributions predicted by both methods, given the same number of stochastic simulations; as shown in Fig. 3b, while at short times, the two are comparable, at long times, the NN-CME's prediction is far more accurate than that of the SSA. Note that training can also be done in steady state, i.e. solving the algebraic equations $\mathbf{A}_\theta(t)\mathbf{P}(t) = 0$; the precision and efficiency of this alternative mode of training the ANN is illustrated and discussed in Fig. S2.

Note that the mapping enabled by the ANN-aided procedure, from delay master equations to NN-CMEs, is also supported by theory for those models which can be solved exactly (see SI Note 5). For Model I, it can be shown that the effective propensity $NN_\theta(n, t)$ is zero for $t < \tau$ and otherwise linear in the nascent copy number $n$ (and independent of time); hence in steady-state conditions, the effective propensity is the same as expected from a first-order degradation process. For Model II, the effective propensity $NN_\theta(n, t)$ is zero for $t < \tau$ and otherwise non-linear in the nascent copy number $n$ (and independent of time). In Fig. 4a we show that for Model II, the effective propensity obtained by the ANN-aided approximation method is in good agreement with the theoretically predicted effective propensity evaluated in steady-state conditions ($t \gg \tau$). The non-linearity of the propensity is an emergent feature of the mapping procedure when transcriptional bursting is present (linear behaviour is

observed for Model I). In Fig. 4b, c, we show how the degree of non-linearity varies with the non-dimensional parameter $\alpha\tau$, which is the ratio of the bursting frequency to the frequency at which nascent RNA gets removed (the elongation rate). The deviations from the conventional linear scaling of the propensity with nascent RNA numbers are manifest when the bursts are produced much slower than the elongation rate. In the inset of Fig. 4b, we show that for hundreds of genes in mouse embryonic stem cells, the value of $\alpha\tau$ is considerably <1 thus showing that the effective degradation propensities for nascent RNA are generally non-linear; often the propensities can be well-approximated by a Hill function (with Hill coefficient <1) over the relevant molecule number range. Since Model II is a good approximation to Model III when gene expression occurs in bursts, it follows that the results shown for Model II also apply to Model III. Note that this also implies that the standard telegraph model of gene expression[4] (equivalent to the NN-CME of Model III with a linear degradation propensity) is not a suitable effective Markovian description for the nascent RNA statistics of most eukaryotic genes.

**Rapid exploration of parameter space and the prediction of a novel type of zero-inflation phenomenon.** Given the computational efficiency of the ANN-aided model approximation, one would expect it to be useful as a means to rapidly explore the phases of a system's behaviour across large swathes of parameter space. This endeavour is only possible if the NN-CME's predictions are accurate across parameter space, which is yet to be seen since we have only shown its accuracy for few parameter sets in Figs. 2 and 3. In what follows, we explicitly verify that the NN-CME can correctly capture all the phases of Model III's behaviour.

We consider the case when the gene spends most of its time in the OFF state (the bursty regime of gene expression). In this case, Model III is well-approximated by Model II (see SI Note 3), and by means of the exact analytical solution of the latter, we identify four regions (I–IV; see Fig. 5a) according to the type of steady-state distribution (see Fig. 5b and its caption for their description). Specifically phase IV is the only region of space where bimodal distributions (peak at zero and at a non-zero value) are found. Theory shows that the conditions (see SI Note 2) that need to be satisfied for this bimodality to manifest are

$$\frac{2 + \frac{2}{b}}{\alpha} < \tau < \frac{b + \frac{1}{b} + 2}{\alpha}, \quad b > 1. \quad (9)$$

By using the NN-CME to randomly sample points in parameter space, we find the same as the theoretical prediction: the distributions are unimodal (dots) except in Region IV where they are bimodal (crosses). Hence this verifies the accuracy of our method across parameter space.

We also note that bimodality in the bursty regime is unexpected because the standard model of gene expression (Model II/III with delayed degradation replaced by first-order degradation[2,5,33]) predicts a unimodal steady-state distribution, which is well-approximated by a negative binomial distribution. Note that Model III is more appropriate to model nascent RNA dynamics than the standard model of gene expression because unlike mature RNA, nascent RNA typically does not get degraded while the RNAP is traversing the gene; rather nascent 'degradation' occurs after a finite elapsed time when it detaches from the gene and becomes mature RNA. Hence Region IV can be understood as delay-induced bimodality or a delay-induced zero-inflation phenomenon. Since there is evidence that the delay time is stochastic rather than fixed[34,35] we also used the NN-CME to

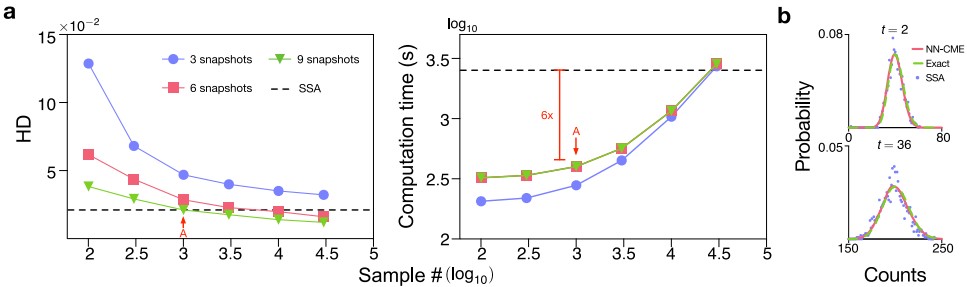

**Fig. 3 Evaluating the performance of the ANN-aided model approximation. a** Precision and computational efficiency of the ANN-aided model approximation as a function of sample size and number of snapshots. The method is benchmarked on Model I since the time-dependent solution of the delay CME is exactly known (see SI Note 1) and hence the accuracy of our method can be precisely quantified. A measure of the accuracy is the average Hellinger distance (HD) between the NN-CME and exact distributions at four different time points. The computation time is equal to the time-to-acquire samples plus time for training. Each data point in the graphs is averaged on three independent trainings. Note that the NN-CME obtained from training with $10^3$ samples produces a distribution that is as precise as that from $3 \times 10^4$ samples using the SSA of the delay CME (shown as a black dashed line); in this case the computation time of the NN-CME is also just 1/6 of the SSA. **b** Comparison of the NN-CME distributions, exact analytical distributions and histograms from stochastic SSA simulations of the delay CME at two different time points; the sampling for both training and the SSA is $10^3$. Note that the NN-CME leads to much more accurate distributions than the SSA for the same number of samples. The rate constants and other parameters related to the ANN's training are specified in SI Table 1.

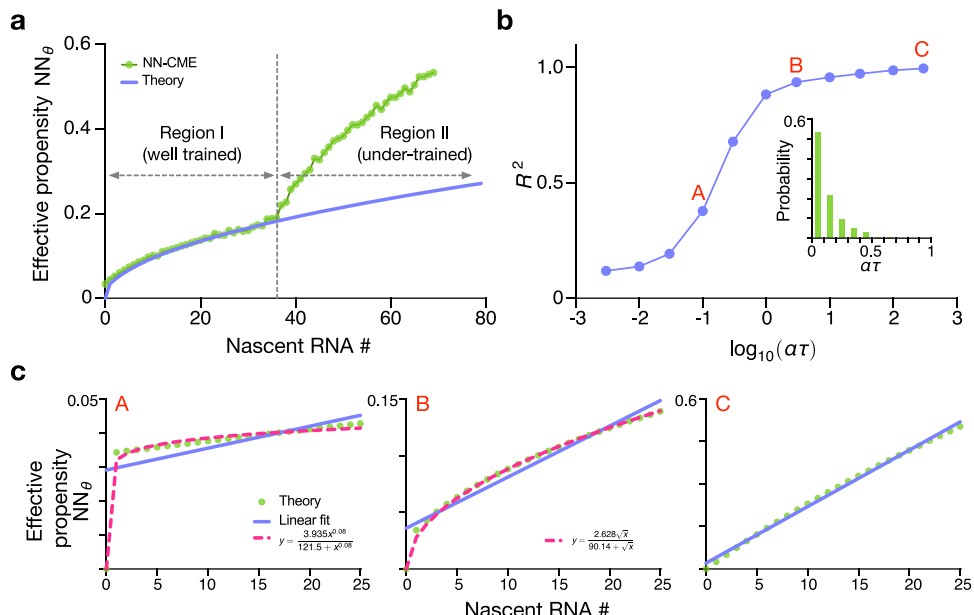

**Fig. 4 Effective degradation propensity of Model II. a** Comparison of the effective degradation propensity $NN_\theta(n)$ in steady-state conditions predicted by theory (solid purple line; Eq. S36 in the SI) and computed by the ANN-aided approximation (green dots). Note that the two agree in Region I where the nascent RNA probability is sufficiently high so that the neural-network coefficients are well-trained. The two are not matched in Region II, since the neural-network coefficients are under trained such that the neural-network output is not reliable. **b** Shows the square of the Pearson correlation coefficient $R^2$ between the effective propensity and the nascent RNA number as a function of the non-dimensional parameter $\alpha\tau$. The non-linearity of the effective propensity rapidly increases as the burst frequency $\alpha$ decreases below the elongation frequency $\tau^{-1}$. Inset shows the histogram of $\alpha\tau$ for 368 genes in mouse embryonic stem cells (see SI Note 6 for details of the histogram). **c** Shows the effective propensity as a function of nascent RNA numbers for points A, B and C labelled in (**b**). The function is almost independent of nascent RNA number for small $\alpha\tau$ (point A), well-approximated by a Hill function of the nascent RNA number for intermediate $\alpha\tau$ (point B), and a linear function of nascent RNA number for large $\alpha\tau$ (point C). Note that the Hill function fits (for points A and B) are only valid over the region shown and break down for larger $n$. The kinetic parameters of Model II are the same as Fig. 2 and the NN-CME is trained at steady state (solving $\mathbf{A}_\theta(t)\mathbf{P}(t) = 0$) using $2 \times 10^5$ samples.

investigate how the nascent RNA distributions change with variance in the delay time when the mean is kept constant: as shown in Fig. 5c, we find that a large increase in the variability of the delay time tends to destroy the peak at zero. Note that for systems with stochastic delay, the training of the ANN-aided approximation remains the same as for those with fixed delay; the advantage of our method is that it can just as easily solve non-Markovian models with stochastic delay as those with

deterministic delay whereas analytically only the latter are amenable to exact solutions when the reaction system is simple enough.

In summary, we find that delayed degradation induces an extra mode peaked at $n = 0$, a type of zero-inflation phenomenon. This phenomenon is commonly seen in single-cell RNA-seq data, and it is usually attributed to the expression drop-off caused by technical noises or sequencing sensitivity[36]. It has also been

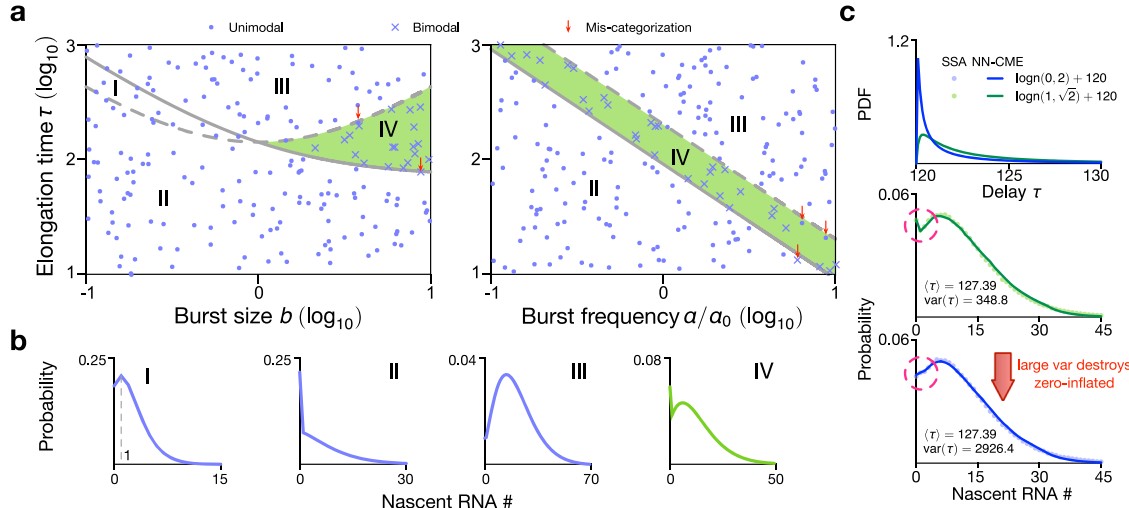

**Fig. 5 Stochastic bifurcation diagram for Model III in the bursty regime ($\sigma_{off} \gg \sigma_{on}$) using the NN-CME and comparison with theory. a** From an analytical approximation of Model III in the bursty regime, the space is divided into four regions according to the type of distributions (shown in **b**): type I, a unimodal distribution with mode = 1; type II, a unimodal distribution with mode = 0; type III, a unimodal distribution with mode > 1; type IV, a bimodal distribution with two modes at zero and a non-zero value. Region IV is highlighted in green since it is a phase that does not exist in the bursty regime of the standard model of gene expression (Model III with delayed degradation replaced by first-order degradation)—this is hence delay-induced bimodality. The lines defining the division of space are: solid line is $(2 + \frac{2}{b})/\alpha$ and the dashed line is $(b + \frac{1}{b} + 2)/\alpha$, which respectively are the lower and upper bounds on $\tau$ given by Eq. (9). To check the accuracy of the ANN-aided model approximation for Model III, we used it to compute the NN-CME and then solved using FSP to obtain nascent number distributions for 200 points in parameter space. These are randomly sampled from the space $\{\rho = 2.11, \sigma_{off} \in 2.11 \times [10^{-1}, 10], \sigma_{on} = 0.0282, \tau \in [10, 10^3]\}$ (left) and $\{\rho = 2.11, \sigma_{off} = 0.609, \sigma_{on} = 0.0282 \times [10^{-1}, 10], \tau \in [10, 10^3]\}$ (right). Dots denote parameter sets for which the NN-CME distributions are unimodal and crosses show those for which the distributions are bimodal. The fact that the vast majority of crosses fall in region IV and the dots outside of it shows that the NN-CME agrees with the analytical approximation of Model III (parameter sets, which mismatch between the NN-CME and theory, are highlighted with red arrows and are very few in number). Note in the left figure of (**a**), the burst frequency is fixed to $\alpha = 0.0282$ (left) while in the right figure, we use $\alpha_0 = 0.0282$ and the burst size is fixed to $b = 3.46$. **c** The NN-CME is learnt from stochastic simulations of the delay model of Model III with the added feature that the elongation time $\tau$ is a random variable sampled from two different lognormal distributions (see top figure). In the middle and bottom figures, we show that the delay-induced bimodality (phase IV) disappears as the variance on the elongation time $\tau$ increases at constant mean. The rate constants and other parameters related to the ANN's training are specified in SI Table 1.

shown[37,38] that it may arise from an extra number of gene states. However our results suggest that delay due to elongation (when the variability in elongation times is small) is another important source contributing to the zero-inflated distributions evident in RNA-seq data.

**Learning the effective master equation of genetic feedback loops from partial abundance data.** Feedback inhibition of gene expression is a common phenomenon in which gene expression is downregulated by its protein product. Given there is sufficient time delay in the regulation process as well as sufficient non-linearity in the mass-action law describing the kinetics of certain reaction steps[39], feedback inhibition can lead to oscillatory gene expression such as that observed in circadian clocks[40].

Here we consider a simple genetic negative feedback loop (see Fig. 6a) whereby (i) a protein $X$ is transcribed by a promoter, (ii) subsequently after a fixed time delay $\tau$, $X$ turns (via some set of unspecified biochemical processes) into a protein $Y$ and (iii) finally $Y$ binds the promoter and reduces the rate of transcription of $X$. Unlike Models I–III considered earlier, the delay master equation corresponding to this model has no known analytical solution. Simulation trajectories verify oscillatory behaviour of this circuit; see Fig. 6b. We use the simulated trajectories of mature protein $Y$ to train the ANN (the objective function only measures the distance between the ANN-predicted distribution of $Y$ and the distribution from stochastic simulations), in a similar way as previous examples (see SI Note 7). In Fig. 6c, d, we show that the time-dependent distributions of both proteins and their

means output using the NN-CME are in excellent agreement with the SSA, even clearly capturing the damped oscillatory behaviour; while for $Y$, this is maybe not so surprising, for $X$, it is remarkable because simulated trajectories of $X$ were not used in the training of the ANN. Hence this shows that the ANN-aided model approximation can learn the effective form of master equations from partial trajectory information, a very useful property if the training data are sparse and available only for some molecular species as commonly the case with experimental data.

**ANN-aided inference of the parameters of bursty transcription.** With a small modification, the ANN-aided model approximation technique besides constructing an approximate NN-CME model, it can also infer the values of kinetic parameters of the data used for training. This is brought about by optimizing not only the weights and biases of the ANN but also simultaneously for the kinetic parameter values. In Fig. 7, we show the results of this method using training data generated by the SSA of Model II with parameters (burst frequency $\alpha$ and size $b$) that have been measured for five mammalian genes[41]. Comparing the latter true parameter values with those obtained from the ANN-aided inference, we conclude that the inference procedure leads to accurate results. Note that the 95% confidence intervals of the estimates are obtained using the profile likelihood method (see SI Note 8 and Fig. 7a, b for an illustration).

We also show that the distribution solution of the NN-CME (which is obtained at one go, together with the inferred parameters) is practically indistinguishable from distributions

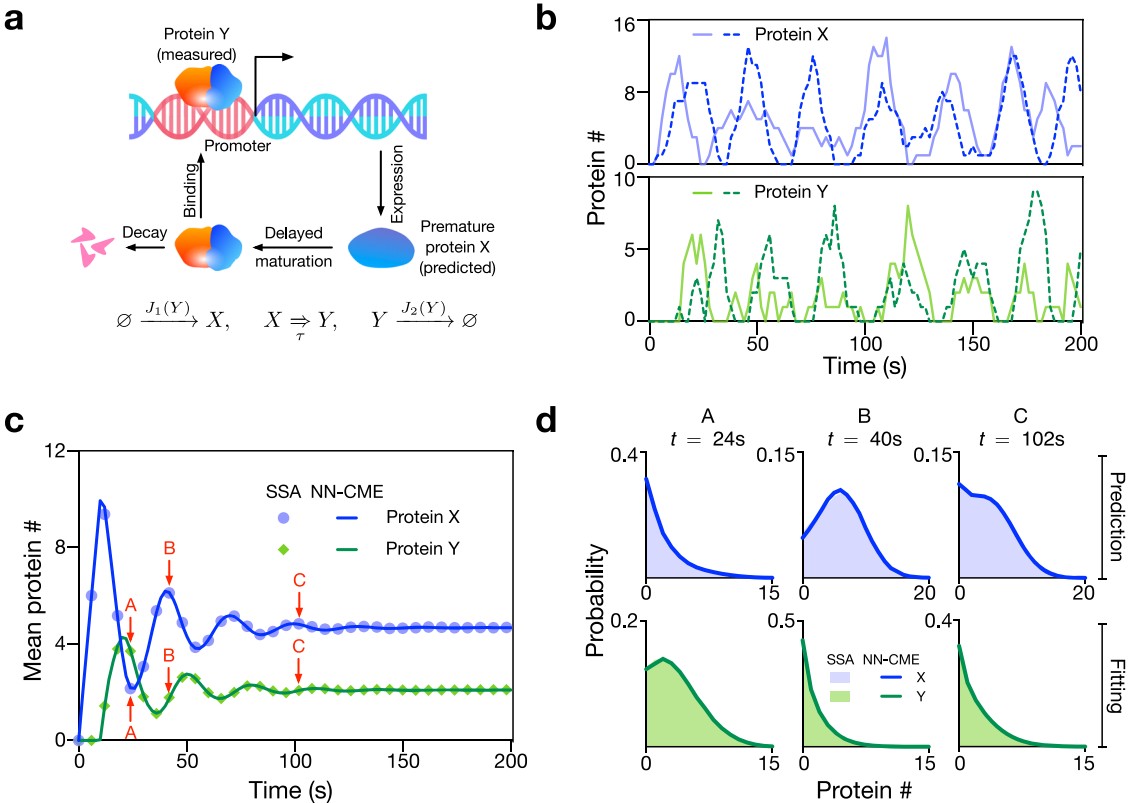

**Fig. 6 NN-CME accurately predicts the properties of a stochastic auto-regulatory model of oscillatory gene expression when only partial data are used for ANN training. a** Illustration of a model of auto-regulation whereby a protein $X$ is transcribed by a gene, then it is transformed after a delay time $\tau$ into a mature protein $Y$, which binds the promoter and represses transcription of $X$. The functions $J_1(Y)$ and $J_2(Y)$ can be found in SI Note 7. **b** Two typical SSA simulations of proteins $X$ and $Y$, clearly showing that single-cell oscillations while noisy, they are sustained. **c, d** The NN-CME is obtained from training the ANN using only protein $Y$ data from SSA simulations of the delay model of the auto-regulatory model. Surprisingly, the NN-CME's solution for the temporal variation of the mean number of both proteins $X$ and $Y$, and for their distributions is in excellent agreement with that of the SSA. Note the distributions in (**d**) are for the three time points labelled A, B and C in (**c**). The rate constants and other parameters related to the ANN's training are specified in SI Table 1.

constructed using the SSA of Model II (the quantile–quantile plots in Fig. 7c are linear with unit slope and zero intercept). In Fig. S3, we show the application of the ANN-aided inference to Model III.

## Discussion

In this paper, we have shown how the training of a three-layer perceptron with a single hidden layer is enough to approximate the delay CME of a non-Markovian model by the NN-CME, which is a master equation with time-dependent propensities (time-inhomogeneous Markov process). Notably, this mapping has been achieved without increasing the effective number of species in the model. Since the NN-CME has no delay terms, it simplifies analysis and simulation; for e.g. the NN-CME can be accurately approximated by a wide range of standard methods[7] and its solution is straightforward using FSP[27]. The method hence enables the efficient study of much more complex non-Markovian models of gene regulation than has been possible to exactly solve analytically or using numerical/simulation methods applied directly to the delay master equation. For example, we showed that our ANN-based method easily solves an extension of Model III where we incorporate noise in the delay time associated with elongation and termination (Fig. 5c), as well as a multi-species model of transcriptional feedback involving delayed maturation of proteins followed by binding to the promoter (Fig. 6). In contrast, the dynamics of these systems cannot be obtained by applying FSP directly to the non-Markovian delay

master equation or using analytical methods reported for non-Markovian stochastic gene expression models[8]. We note that while neural networks have been recently used to approximate partial differential equations in physics, chemistry and biology, to our knowledge, our work represents their first use in approximating equations describing the time-evolution of stochastic processes in continuous time and with a discrete state space, e.g. systems describing cellular biochemistry where the discreteness is an important feature of the system due to the low copy number of DNA and mRNA molecules involved[42].

We find that to obtain an accurate NN-CME, training only needs a small sample size (of the order of a thousand SSA trajectories which is computationally cheap), it can be done with partial data (some species data can be missing) and simultaneously one can obtain estimates of the kinetic parameters. The latter is particularly relevant if the training data are collected experimentally, e.g. by measuring nascent RNA numbers using live-cell imaging techniques (such as the MS2 system) at several time points for many cells[43,44]. Our ANN-based inference method rests upon the matching of distributions and hence similarly to non-ANN-based methods developed in refs. [45,46], it avoids the pitfalls of moment-based inference[47,48]. We note that the vast majority of existing inference methods are for stochastic systems with no delayed reactions; a notable exception is ref. [49]. We also note that the ability to approximate solutions of delay master equations from simulated data while simultaneously optimizing for the parameters has not been demonstrated before; deep learning frameworks have previously achieved similar feats for deterministic

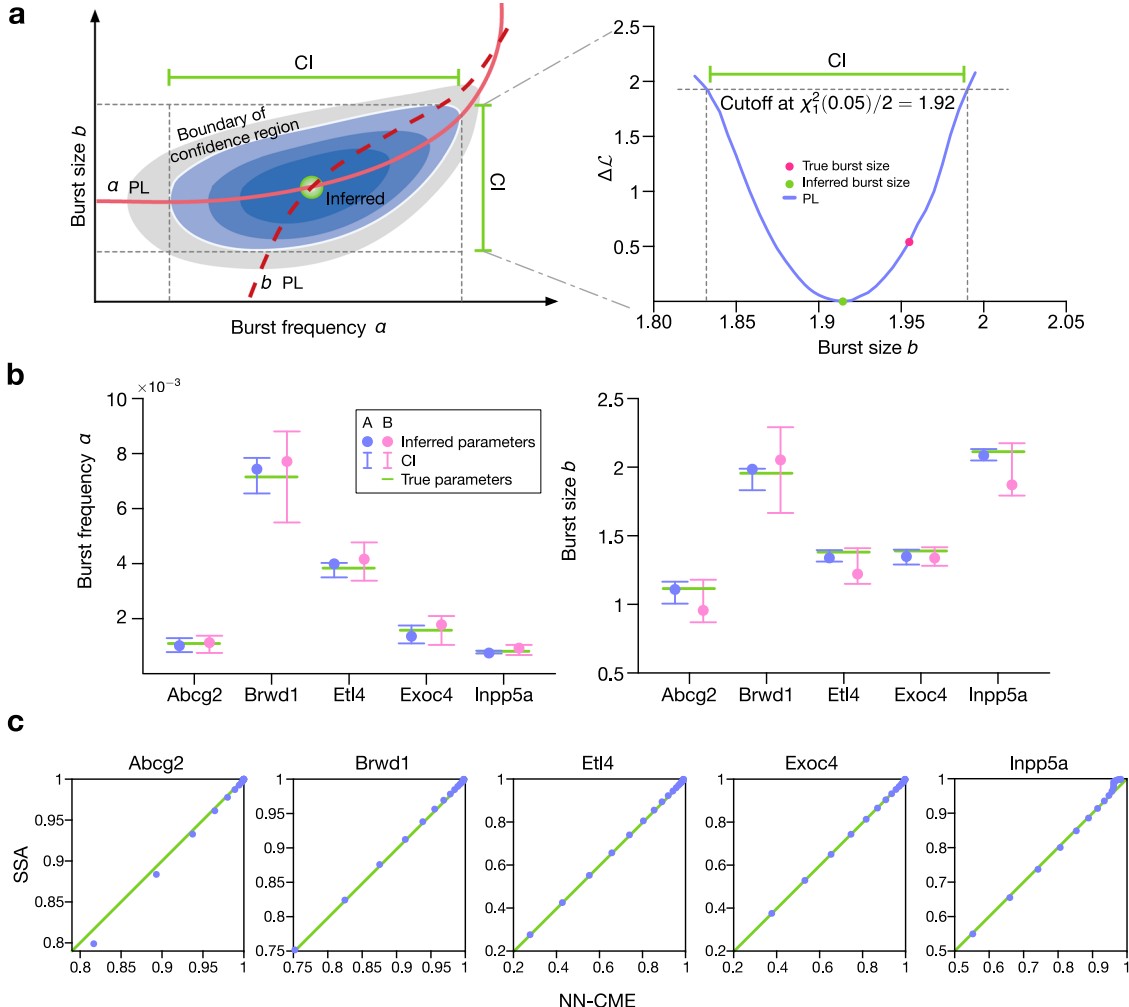

**Fig. 7 ANN-aided model approximation seamlessly integrates the inference of kinetic parameters and approximation of the delay CME by a NN-CME.**
The unknown kinetic parameters can be treated in the same way as neural-network coefficients (weight and biases) and optimized to minimize the objective function. Application to Model II. **a** Sketch of the computation of the 95% confidence interval (CI) of the inferred kinetic parameters. Blue areas indicate the 95% confidence region, while the grey area shows the non-confidence region. Both solid and dashed red lines show the profile likelihoods (PLs) of burst frequency $\alpha$ and burst size $b$, respectively (See SI Note 8 for details). **b** Inferred values of $\alpha$ and burst size $b$ (dots), their 95% CIs (error bars) and the true values (green lines) for five mammalian genes. Inference by using ANN-aided model approximation is robust against size of dataset: Dataset A (blue, 100 snapshots and $10^4$ cells) and Dataset B (red, 50 snapshots and $10^3$ cells) produce similar results. **c** Quantile–quantile plots for the steady-state distributions of the NN-CME and those obtained from the SSA; the linearity confirms that the ANN-aided model approximation can accurately approximate the distribution using the NN-CME even when the optimization is over both the kinetic parameters and the neural-network coefficients. The rate constants and other parameters related to the ANN's training are specified in SI Table 1.

models[18,21] and more recently for stochastic models described by multi-dimensional Fokker–Planck equations[50,51].

The ANN-based procedure described in this article is most useful to learn effective propensities of those biomolecular processes which we don't know how to model well using a Markovian approach. The input data for the ANN's training can be experimental data or that generated by a complex model. The complex model could be non-Markovian as in this paper or else could be a Markovian model with many more species and reactions than the effective Markovian model that the ANN is trying to learn. In some cases the procedure will show that a mapping is not possible. For example, here we have shown that the standard telegraph model of gene expression (equivalent to the NN-CME of Model III with a linear degradation propensity) is not a suitable Markovian description for the nascent RNA dynamics of most eukaryotic genes (it is typically a good description for mature RNA dynamics as has been analytically shown in ref. [9]).

Recent work has shown that differential equation models describing the time-evolution of average agent density can be learnt (using sparse regression) from agent-based model simulations of spatial reaction-diffusion processes[52]. Such models can describe for e.g. intracellular biochemical processes in crowded conditions[53] or multi-scale tissue dynamics including cell movement, growth, adhesion, death, mitosis and chemotaxis[54–56]. Some of these models have been shown to display non-Markovian behaviour[57]. While here we showcased the ANN-based method using non-Markovian delay CMEs, one could also use for training, data generated by spatially resolved particle-based simulations, as the examples above. The application of our method would provide a master equation that effectively captures stochastic dynamics at the population level of description and avoids the pitfalls of commonly used analytical approximation methods, e.g. mean-field approximations, to obtain reduced stochastic descriptions.

**Reporting summary**. Further information on research design is available in the Nature Research Reporting Summary linked to this article.

## Data availability

The experimental data shown as an inset in Fig. 4b can be found at https://doi.org/10.5281/zenodo.4643094.

## Code availability

The codes, readme file and data for ANN-aided model approximation can be found at https://doi.org/10.5281/zenodo.4643094. The codes are implemented by `Julia` 1.4.2 and its package `Flux` v0.10.4, `DifferentialEquations` v6.15.0 and `DiffEqSensitivity` v6.26.0.

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

## Acknowledgements

Z.C., W.D. and F.Q. acknowledge the support from Natural Science Foundation of China (NSFC No. 61988101); Z.C. acknowledges the support from NSFC No. 62073137; W.D. acknowledges the support from NSFC No. 61725301; Q.J. and S.Y. acknowledge the support from NSFC No. 61973119, National Key Research and Development Program of China (2020YFA0908303) and Shanghai Rising-Star Program (20QA1402600); R.G. thanks the support from the Leverhulme Trust Grant (RPG-2018-423). We thank James Holehouse, Kaan Öcal and Guido Sanguinetti for useful discussions and insightful feedback.

## Author contributions

Z.C. and R.G designed research, supervised research, acquired funding and wrote the manuscript with input from the others. Q.J., X.F. and S.Y. performed research, analysed the data and wrote the manuscript. W.D. and F.Q. analysed the data and acquired funding. R.L. was involved in data analysis and graphical illustration.

## Competing interests

The authors declare no competing interests.
