## [Peer Review File · Nature Communications]

Reviewers' Comments:

Reviewer #1:

Remarks to the Author:

The authors accomplished two goals in this paper. First, in the supplement, the authors derived the basic time-delayed model based on the theory of conditional probability, and model II and III by incorporating more details in the transcriptional process of binding of RNAP's to the promoter. Secondly, the authors used the non-homogeneous Markov-chain to model/approximate the time-delay model. This was achieved by data training from neural network, one of the highlight of this paper. The authors validated this approach by providing computational evidence that the Markov model can closely approximate the simulated time-delay model, surprisingly, even with partial abundance data. What is missing and desirable, is some theoretical validation of this approach. In particular, the authors switched from Master equation 1 to master equation 2 without much elaboration. Can you provide an argument or reference for equation 2, just like you did for equation 1? The particular form of the transition matrix on page 4 is important later on. This needs some justification, perhaps a diagram will help.

Reviewer #2:

Remarks to the Author:

Review: Jiang et al.

The authors present a method for using artificial neural network to derive approximations for non-Markovian models of mRNA kinetics. Older models for these kinetics were discrete and memoryless (Markovian), reflecting the limitations of available experimental data. However, as the resolution and accuracy of experimental methods have improved, one can now quantify the amount of nascent (actively transcribed) mRNA at individual gene loci. To capture this type of data, theoretical models need to incorporate the process of transcription elongation (and possibly, degradation), thus introducing temporal memory and losing the Markovian property. Consequently, the available tools for solving the corresponding master equation, such as the finite state projection method, cannot be used anymore. Here, the authors describe a neural network-based approximation approach to map these non-Markovian models back to simple Markovian ones, thus allowing again the application of these tools. I found the work to be very exciting, with great promise of advancing our ability to model gene regulation with better realism. I thus support its publication in Nature Communications.

Before publication, I ask the authors to address a number of points:

- 1) The non-discreteness of mRNA numbers: Current methods for measuring mRNA, and the corresponding models, allow for the fact that mRNA numbers are not discrete but rather continuous: Most mRNA molecules are fractional (incomplete), due to the finite speeds of mRNA elongation and degradation (see, e.g., PMID: 25964259, 31527794). The current work still describes mRNA as discrete. Thus, if I understand it, the species modeled is not actually the amount of nascent mRNA but rather the number of RNAPs on the gene. Is that correct?
- 2) Following on the point above, a little more should be said about how other researchers have modeled (and solved) the non-Markovian and continuous kinetics of nascent (and mature) mRNA. What are the pros and cons of these previous approaches versus the current one?
- 3) Interpretation: What can we learn from the results of the neural network, and how do we interpret the "learned" network? This will be critical when the authors' method is used for building models for real experimental data: One may obtain a good fit for the data, but that would only be useful if they can then interpret what the successful model reveals about the underlying kinetics.

4) Title: I feel that the current title buries the lede. The key innovation of the new method is its ability to handle non-Markovian models. This should be made clear from the title.

Response letter for “Neural network aided approximation and parameter inference of non-Markovian models of gene expression”

Qingchao Jiang, Xiaoming Fu, Shifu Yan, Runlai Li, Wenli Du, Zhixing Cao, Feng Qian, and Ramon Grima

*East China University of Science and Technology
The University of Edinburgh*

February 19, 2021

Reviewer 1

The authors accomplished two goals in this paper. First, in the supplement, the authors derived the basic time-delayed model based on the theory of conditional probability, and model II and III by incorporating more details in the transcriptional process of binding of RNAP's to the promoter. Secondly, the authors used the non-homogeneous Markov-chain to model/approximate the time-delay model. This was achieved by data training from neural network, one of the highlight of this paper. The authors validated this approach by providing computational evidence that the Markov model can closely approximate the simulated time-delay model, surprisingly, even with partial abundance data.

Response: We thank Reviewer 1 for the positive comments about our work.

1. What is missing and desirable, is some theoretical validation of this approach. In particular, the authors switched from Master equation 1 to master equation 2 without much elaboration. Can you provide an argument or reference for equation 2, just like you did for equation 1?

Response: Inspired by the referee's comment, using the exact solutions of Models I and II, we have now shown theoretically that indeed it is possible to map a delay master equation (the non-Markovian model) on to a master equation with an effective degradation propensity (the Markovian model). The derivations can be found in the new SI Note 5 and a discussion can be found in the main text on P. 9-10. We have also added a new Section 2.1 in the SI to show the derivation of the delay master equation for Model II (before we only had it for Models I and III).

2. The particular form of the transition matrix on page 4 is important later on. This needs some justification, perhaps a diagram will help.

Response: We have now justified the form of the transition matrix by adding a discussion of the intermediates steps necessary to go from Eq. (3) to Eq. (4). This can be found on P. 3-4. We note that these steps are none other than those implicit in the finite state projection (FSP) method.

Reviewer 2

The authors present a method for using artificial neural network to derive approximations for non-Markovian models of mRNA kinetics. Older models for these kinetics were discrete and memoryless (Markovian), reflecting the limitations of available experimental data. However, as the resolution and accuracy of experimental methods have improved, one can now quantify the amount of nascent (actively transcribed) mRNA at individual gene loci. To capture this type of data, theoretical models need to incorporate the process of transcription elongation (and possibly, degradation), thus introducing temporal memory and losing the Markovian property. Consequently, the available tools for solving the corresponding master equation, such as the finite state projection method, cannot be used anymore. Here, the authors describe a neural network-based approximation approach to map these non-Markovian models back to simple Markovian ones, thus allowing again the application of these tools. I found the work to be very exciting, with great promise of advancing our ability to model gene regulation with better realism. I thus support its publication in Nature Communications.

Response: We thank Reviewer 2 for the positive comments about our work.

1. The non-discreteness of mRNA numbers: Current methods for measuring mRNA, and the corresponding models, allow for the fact that mRNA numbers are not discrete but rather continuous: Most mRNA molecules are fractional (incomplete), due to the finite speeds of mRNA elongation and degradation (see, e.g., PMID: 25964259, 31527794). The current work still describes mRNA as discrete. Thus, if I understand it, the species modeled is not actually the amount of nascent mRNA but rather the number of RNAPs on the gene. Is that correct?

Response: We full agree with the referee. We have now clarified this point after we introduce Model I on P. 3. Specifically we state “Note that the number of RNAPs bound to the gene is equal to the number of nascent RNA molecules present, irrespective of their lengths [22] (for an illustration see Fig. 2a Model I). We note that the signal from single-molecule fluorescence in situ hybridization (smFISH) probes corresponds to measuring the total length of nascent RNA, summed over multiple molecules present at the gene; thus the number of nascent RNA estimated from such experiments would lead to a continuous rather than a discrete number [7, 23, 24]. Our present formulation ignores the complexities introduced by smFISH and is rather compatible with experiments that can directly quantify the number of RNAPs bound to a gene [25]”.

2. Following on the point above, a little more should be said about how other researchers have modeled (and solved) the non-Markovian and continuous kinetics of nascent (and mature) mRNA. What are the pros and cons of these previous approaches versus the current one?

Response: We have now introduced sentences after the introduction of Models I and and III (specifically after Eqs. 2 and 8) citing the literature reporting an exact solution of the delay master equations corresponding to these models. Note that to our knowledge Model II has not been exactly solved before (we report an exact solution in the SI Note 2). On P. 7 we have discussed how Model III was solved in Ref. 7 which presented the first analytical solution of a delay model of transcription dynamics and that considers different cases, some in which the nascent RNA is discrete (which would correspond to our present case of equating nascent RNA with the number of bound RNAPs) and other cases where the nascent RNA is a continuous number (due to taking into account the length of the nascent RNA bound to the RNAP). We have also clarified by a sentence in the Introduction that modelling of mature RNA has been done using a Markovian model, i.e. the famous telegraph or two-state model.

As to contrasting previous approaches versus the current one, the major advantage of our approach is that it is not limited by the complexity of the model whereas the analytical machinery of previous approaches cannot be pushed beyond that of simple models (such as Models I-III). Of course, the dis-

advantage of our methodology is that it cannot give analytical expressions which can help intuition and interpretation. We clarified this by a paragraph in the Discussion: “The method hence enables the efficient study of much more complex non-Markovian models of gene regulation than has been possible to exactly solve analytically or using numerical/simulation methods applied directly to the delay master equation. For example we showed that our ANN-based method easily solves an extension of Model III where we incorporate noise in the delay time associated with elongation and termination (Fig. 5c), as well as a multi-species model of transcriptional feedback involving delayed maturation of proteins followed by binding to the promoter (Fig. 6). In contrast, the dynamics of these systems cannot be obtained by applying FSP directly to the non-Markovian delay master equation or using analytical methods reported for non-Markovian stochastic gene expression models [7].”

3. Interpretation: What can we learn from the results of the neural network, and how do we interpret the “learned” network? This will be critical when the authors’ method is used for building models for real experimental data: One may obtain a good fit for the data, but that would only be useful if they can then interpret what the successful model reveals about the underlying kinetics.

Response: This is an excellent question and indeed one that can be posed to any neural network based study. Essentially in our case, the neural network is used to answer the following question: what is the effective degradation propensity of nascent RNA in a chemical master equation (Markovian) model such that the solution of this matches distributions of copy numbers determined experimentally or else using a non-Markovian model. Previously we showed that such a mapping between the two types of models is possible but we did not show the form of the effective propensity learnt by the ANN-based procedure. We have now added a new figure (Fig. 4) and added discussion on P. 9-10 discussing the learnt propensity for Model II. Note that we focus on Model II because it a good approximation to Model III when gene expression occurs in bursts. In Fig. 4a we show that for Model II, the effective propensity obtained by the ANN-aided approximation method is generally nonlinear in the nascent copy number. Note that for a conventional first-order process, the degradation propensity is linear in n . This non-linearity of the effective propensity is an emergent feature of the mapping procedure (from delay to chemical master equations) when transcriptional bursting is present (the effective degradation propensity is linear for Model I where there is no bursty expression). In Figs. 4b,c, we show how the degree of non-linearity varies with the nondimensional parameter $\alpha\tau$ which is the ratio of the bursting frequency to the frequency at which nascent RNA gets removed (the elongation rate). The deviations from the conventional linear scaling of the propensity with nascent RNA numbers are manifest when the bursts are produced much slower than the elongation rate. In the inset of Fig. 4b we show that for hundreds of genes in mouse embryonic stem cells, the value of $\alpha\tau$ is considerably less than 1 thus showing that the effective degradation propensities for nascent RNA are generally nonlinear; in some cases these can be well approximated by a Hill function. Note that this also implies that the standard telegraph model of gene expression (equivalent to the NN-CME of Model III with a linear degradation propensity) is not a suitable Markovian description for most eukaryotic genes. The ANN-based procedure is hence most useful to learn effective propensities of those biomolecular processes which we don’t know how to model well using a Markovian approach. We have elaborated on these points in a new paragraph in the Discussion.

4. Title: I feel that the current title buries the lede. The key innovation of the new method is its ability to handle non-Markovian models. This should be made clear from the title.

Response: We have changed the word “stochastic” in the title to “non-Markovian” to better reflect the novelty of the paper.

Reviewers' Comments:

Reviewer #1:

Remarks to the Author:

The authors seemed to have addressed my questions adequately. I recommend this paper to be published.

Reviewer #2:

Remarks to the Author:

The authors have addressed my concerns in a satisfactory manner.